# In Vitro and In Vivo Studies of Oritavancin and Fosfomycin Synergism against Vancomycin-Resistant *Enterococcus faecium*

**DOI:** 10.3390/antibiotics11101334

**Published:** 2022-09-29

**Authors:** Cristina Lagatolla, Jai W. Mehat, Roberto Marcello La Ragione, Roberto Luzzati, Stefano Di Bella

**Affiliations:** 1Department of Life Sciences, Trieste University, 34127 Trieste, Italy; 2School of Biosciences and Medicine, University of Surrey, Guildford GU2 7XH, UK; 3School of Veterinary Medicine, Faculty of Health and Medical Sciences, University of Surrey, Guildford GU2 7XH, UK; 4Clinical Department of Medical, Surgical and Health Sciences, Trieste University, 34127 Trieste, Italy

**Keywords:** vancomycin-resistant enterococci, *Enterococcus faecium*, synergy, synergism, oritavancin, fosfomycin, in vivo, *Galleria mellonella*, combination

## Abstract

Therapeutic options for infections caused by vancomycin-resistant enterococci are currently suboptimal. Combination regimens where fosfomycin is used alongside existing treatments are emerging given the proven synergistic potential and PK/PD properties. In the studies presented here, we tested five *vanA* and five *vanB* clinical isolates of *Enterococcus faecium* using a combination of oritavancin + fosfomycin both in vitro (checkerboard, time killing) and in vivo (*Galleria mellonella*). The combination of oritavancin and fosfomycin increased drug susceptibility, showing a synergistic effect in 80% of isolates and an additive effect in the remaining isolates. The combination restored fosfomycin susceptibility in 85% of fosfomycin-resistant isolates. Time killing on four selected isolates demonstrated that the combination of oritavancin and fosfomycin provided a CFU/mL reduction > 2 log_10_ compared with the most effective drug alone and prevented the bacterial regrowth seen after 8–24 h at sub-inhibitory drug concentrations. In addition, the combination was also tested in a biofilm assay with two isolates, and a strong synergistic effect was observed in one isolate and an additive effect in the other. Finally, we demonstrated in vivo (*Galleria mellonella*) a higher survival rate of the larvae treated with the combination therapy compared to monotherapy (fosfomycin or oritavancin alone). Our study provides preclinical evidence to support trials combining oritavancin and fosfomycin for VRE BSI in humans, even when biofilm is involved.

## 1. Introduction

The treatment of infections caused by vancomycin-resistant enterococci (VRE) is a challenge for clinicians [1]. Therapeutic options are limited and are often suboptimal. Commonly used treatment options include linezolid and daptomycin; however, there are a number of limitations associated with their use. Firstly, linezolid is a bacteriostatic lipophilic drug that is ill-suited for bloodstream infections (BSI); secondly, daptomycin, even at doses of 10–12 mg/kg/day, does not reach adequate concentrations to treat infections caused by enterococci, with an MIC of 4–8 mg/L [2,3].

When considering enterococcal infections, it is important to consider the adhesive potential of enterococci, which provides them with good tropism for heart and medical devices (e.g., cardiac valves, vascular prosthesis, urinary stents, etc.) in infections [4]. This is important because, in these circumstances, we should ideally administer a therapy with anti-biofilm properties. Biofilms confer tolerance to the immune system, as well as tolerance and resistance to antimicrobials, mainly through the extracellular polymeric matrix and the change in the metabolism of embedded bacteria (growth reduction) [5,6].

To summarise, the difficulty of treating VRE infections may depend on several variables: (1) the site of infection (e.g., issues with lipophilic antibiotics); (2) the inoculum size (e.g., intra-abdominal abscesses); (3) the presence of biofilm (e.g., vascular prosthesis infections); (4) the resistance profile of the microorganism (e.g., high daptomycin MIC).

In view of these issues, the scientific community needs not only “real-life” data on new anti-VRE drugs such as eravacycline, omadacycline and oritavancin, but also data from combination therapies that could potentially allow us to: (1) reduce the MIC of co-administered antibacterials; (2) ameliorate pharmacokinetic/pharmacodynamic (PK/PD) parameters (e.g., associating a hydrophilic with a lipophilic drug) and (3) provide increased activity against biofilm-embedded bacteria.

From this perspective, we considered the combination between oritavancin and fosfomycin, an appealing one. 

Oritavancin is a semisynthetic lipoglycopeptide with in vitro activity against both vancomycin-susceptible and -resistant enterococci [7]. It inhibits peptidoglycan at two stages and has an excellent MIC against VRE. However, it is a lipophilic drug; therefore, technically, it would not be an optimal choice for bloodstream infections if administered alone.

Fosfomycin is an old drug but is being increasingly used in combination regimens with several partner drugs [8]. It is a small hydrophilic drug with synergistic properties with several other antibiotics (e.g., daptomycin and rifampin against *Enterococcus*) [9]; it has good in vitro activity against VRE [10] and good biofilm penetration [11]. These properties make fosfomycin a good partner for oritavancin. 

We assessed the combination of oritavancin + fosfomycin against vancomycin-resistant *E. faecium* clinical strains isolated from BSI patients using in vitro assays and in vivo models (*Galleria mellonella*).

## 2. Methods

### 2.1. Bacterial Strains and Culture Conditions

Five vanA and five vanB clinical isolates of *Enterococcus faecium* (Ef), stored at −80 °C and previously described [12], were streaked onto brain/heart infusion (BHI) agar and routinely cultured in BHI broth at 37 °C for 20 h. Microbial identification and antimicrobial susceptibility testing to vancomycin and teicoplanin were performed using the VITEK2 automated system (bioMèrieux, Marcy-L’Etoile, France), and confirmatory minimum inhibitory concentration (MIC) testing was carried out by the broth dilution method, following standard criteria [13]. Results were interpreted according to the European Committee on Antimicrobial Susceptibility Testing (EUCAST)-approved breakpoints [14]. Susceptibility testing for fosfomycin was preliminary performed through the agar dilution method using the AD Fosfomycin 0.25–256 Kit (Liofilchem S.r.l., Roseto degli Abruzzi, Italy). MIC assays were also performed in Cation-Adjusted Müller–Hinton Broth (CAMHB) (Oxoid, Cheshire, United Kingdom), both to confirm the fosfomycin MIC and to evaluate oritavancin susceptibility, since the most pragmatical method to carry out checkerboard and time-kill assays is in liquid medium. Oritavancin and fosfomycin were purchased from Merck Life Science (Milan, Italy) in powder form and dissolved in deionised water, at a concentration of 1 mg/mL and 40.0 mg/mL, respectively. Polysorbate-80 0.002% was added to oritavancin solution, according to the Clinical and Laboratory Standards Institute guidelines [13]. Both stock solutions were stored at −20 °C and diluted in CAMHB for each experiment. Polysorbate-80 0.002% and glucose-6-phosphate (G6P) 25 mg/L were added to CAMHB, as required for in vitro testing of oritavancin and fosfomycin, respectively. *E. faecalis* ATCC 29212 was used as a control in the antimicrobial susceptibility assays.

### 2.2. Synergy Testing by Checkerboard Assay

The activity of fosfomycin and oritavancin in combination was assessed using a checkerboard assay using a 7-by-5 well configuration. In a 96-well plate, the two drugs were placed at increasing concentrations. Then, 25 μL serial dilutions containing four times the desired concentrations of each drug were inoculated in the 96-well plate, to obtain different combinations of the two drugs. Next, 50 μL of the bacterial culture diluted overnight, containing 10^6^ CFU/mL, was added to obtain an initial inoculum of 5 × 10^5^ CFU/mL. Values of MICs of the drugs alone and in combination were determined as the lowest drug concentrations inhibiting bacterial growth after overnight incubation at 37 °C. Every checkerboard assay was repeated at least twice in triplicate to confirm the results. Interactions between antibiotics were determined by calculating the fractional inhibitory concentration index (FICI), according to the following formula: MIC of drug A in combination/MIC of drug A acting alone + MIC of drug B in combination/MIC of drug B acting alone. Results were interpreted as follows: FICI ≤ 0.5, synergism; 0.5 < FICI ≤ 1, additive effect; 1 < FICI ≤ 4, no interaction; and FICI > 4.0, antagonism [15].

### 2.3. Time-Kill Assay

Two vanA and two vanB isolates resistant to fosfomycin and against which the combination fosfomycin + oritavancin showed a synergistic effect were tested using a time-kill assay. For each isolate, time-kill curves were evaluated in CAMHB containing: fosfomycin MIC, oritavancin MIC, the combination of the two drugs at concentrations that showed effectiveness in the checkerboard assay (henceforth named subMIC), fosfomycin subMIC and oritavancin subMIC. The assay was carried out in 2 mL broth samples inoculated in a 24-well plate on an initial inoculum of 5 × 10^5^ CFU/mL. A positive control in CAMHB without drug was always included. Viable cells were evaluated by plating serial dilutions at 0, 4, 8 and 24 h. Time–kill curves were generated by plotting mean colony counts of three independent experiments (log_10_ CFU/mL) ± standard deviation. Bactericidal activity was defined as a 3 log_10_ CFU/mL reduction from baseline at 24 h. Synergy between two agents was defined as a 2log_10_ CFU/mL reduction at 24 h compared with the most active agent alone [15].

### 2.4. Biofilm Assays

Biofilm (BF) production was initially evaluated both in BHI and in BHI supplemented with 5% defibrinated sheep blood (Blood-BHI). Each strain was cultured overnight in BHI at 37 °C with orbital shaking at 100 rpm, diluted 1000× both in BHI and in B-BHI and inoculated in a 96-well polystyrene microtiter plate (200 μL/well, carried out in triplicate). Negative controls (culture medium without bacteria) were included in all assays. Following 24 h of incubation with orbital shaking at 100 rpm, the culture suspensions were removed, and each well was then gently rinsed with 200 μL of sterile normal saline. After a further 1 h of incubation at 60 °C to fix the BF, each well was stained with 200 μL of crystal violet 2% (CV) (Sigma, Saint Louis, MO, USA) for 15 min, rinsed with water and dried at room temperature. The amount of CV bound to BF was evaluated by measuring the OD at 570 nm after solubilisation in 200 μL of 33% acetic acid for 30 min. The mean values of the negative control samples were subtracted from the test samples, to eliminate possible interference of the medium. Final results were the means of three independent experiments ± standard deviation.

The antimicrobial activity of fosfomycin and oritavancin, alone and in combination, on sessile cells was evaluated using a BF susceptibility assay [16], with minor modifications. Briefly, the BF grown on peg lids immersed in 100 μL of bacterial suspension grown in B-BHI was incubated in CAMHB containing Polysorbate-80 0.002% and G6P 25 mg/L supplemented with serial dilutions of the drugs; after 24 h at 37 °C with orbital shaking at 100 rpm, it was rinsed three times with sterile saline, transferred to fresh BHI by centrifugation at 850× *g* for 20 min and further incubated for 6 h. The biofilm inhibitory concentration (BIC) was established as the lowest concentration that inhibited the growth of detached cells (OD590 < 0.05).

### 2.5. In Vivo Toxicity in Galleria mellonella

In vivo toxicity of oritavancin and fosfomycin, individually and in combination, was determined using the *G. mellonella* infection model, as previously described [17].

*G. mellonella* were purchased from LiveFoods UK Ltd. and stored at 15 °C prior to use. Only larvae weighing between 0.7 g and 1.3 g and showing no discolouration or injury were used for the assays. Groups of 10 larvae were injected into the top left proleg with either fosfomycin (512 µg/mL), oritavancin (0. 512 µg/mL) or a combination of both agents. These concentrations were equivalent to the highest MIC observed for each agent against *E. faecium* isolates. Larvae inoculated with PBS were used as injection controls. All larvae were incubated aerobically at 37 °C and assessed for mortality at 24-h intervals. Larvae were classed as dead when an absence of movement in response to stimuli was observed. All assays were performed in triplicate and the data were pooled and plotted using GraphPad Prism 8.4.3 software (GraphPad Software Inc., San Diego, CA, USA).

### 2.6. G. mellonella Treatment Assay Methods

Sixteen-hour cultures of Ef-3, Ef-4, Ef-8 and Ef-10, grown aerobically in BHI at 37 °C, were washed in PBS prior to the inoculation of larvae. Viable counts of each strain were determined by plating dilutions onto Müller–Hinton agar and incubating them aerobically at 37 °C for 24 h. Groups of 10 larvae were infected via the top right proleg with either Ef-3 (5 × 10^4^ CFU/Larvae), Ef-4 (6 × 10^4^ CFU/Larvae), Ef-8 (1 × 10^5^ CFU/Larvae) or Ef-10 (2 × 10^4^ CFU/Larvae), or inoculated with 10 µL PBS. Inocula concentration was chosen based on the ability to produce staggered mortality over 72 h. Within 15 min of infection, a second injection into the left proleg was performed to administer fosfomycin, oritavancin, a combination of both agents or PBS control. The concentration of each drug was equal to that used in combination in time-kill assays. Larvae were incubated aerobically at 37 °C and were scored for mortality at 0, 24, 48 and 72 h post-infection. All assays were performed in triplicate and the data were pooled and plotted using GraphPad Prism 8.4.3 software (GraphPad Software Inc., San Diego, CA, USA). Statistical survival analysis was performed using a Mantel–Cox log-rank test.

## 3. Results

### 3.1. Synergism between Fosfomycin and Oritavancin: In Vitro Analysis

Preliminary evaluation of the oritavancin susceptibility of the ten isolates tested in this study revealed quite high MIC values. Indeed, even if it is not possible to categorise these strains as susceptible or resistant, since the breakpoint values for VRE strains have not yet been defined, it is noteworthy that all but one demonstrated an oritavancin MIC of 0.256 or 0.512 µg/mL, which is higher than the breakpoints defined for vancomycin-susceptible *E. faecalis* and for *Staphylococcus aureus*. The MICs of fosfomycin described in a previous study [12] were confirmed in the checkerboard assay, which was initially used to evaluate the activity of the FOS + ORI combination. In all isolates, the combination increased drug susceptibility, showing a synergistic effect in eight out of ten isolates (FICI values ranging from 0.133 to 0.375) and an additive effect in the other two (FICI values of 0.5 and 0.531). Interestingly, the combination restored fosfomycin susceptibility in six out of seven fosfomycin-resistant isolates; in the remaining isolate (Ef-8), MICFOS was decreased to 128 µg/mL, which is categorised as intermediate (Table 1).

Synergism between fosfomycin and oritavancin was also confirmed by a time-kill assay performed on four isolates (two carrying the *vanA* determinant and two carrying the *vanB* determinant) at concentrations of the drugs that were effective in the checkerboard assay. The time-kill graphs shown in Figure 1 clearly indicate that, for all isolates, the drug combination caused a reduction in CFU/mL of more than 2 log_10_ (e.g., between 2.5 and 3.4 log_10_) compared with the most effective drug alone. The combination showed a constant bacteriostatic effect throughout the 24-h incubation, in contrast to the trend observed with the two drugs used alone, which showed an initial bactericidal effect, but was followed by regrowth between 8 and 24 h.

### 3.2. Biofilm Assays

The ability of Ef isolates to form BF was investigated in advance, by crystal violet staining, with bacteria cultured in both BHI and Blood-BHI. Figure 2 clearly shows the higher amount of BF produced by isolates grown in the blood-containing medium, which was 5 to 14 times more abundant compared to bacteria grown in BHI without blood. 

The activity of fosfomycin and oritavancin, alone and in combination, on sessile cells from two representative isolates was also evaluated by performing a BIC assay on Ef-3 (*vanB*) and Ef-10 (*vanA*). As shown in Table 2, the sessile cells of these isolates were always more resistant to both drugs than the planktonic cells, with BIC values at least four times higher than the MIC for the two drugs tested alone. When the drugs were tested in combination, the two isolates showed different behavior. A strong synergistic effect was observed for Ef-3, with BIC values lowered 32-fold for fosfomycin and 16-fold for oritavancin, resulting in a FICI value of 0.094. On the Ef-10 biofilm, however, the combination showed an additive effect, with a FICI value of 0.563. In this case, we observed a good reduction in the BIC of fosfomycin (>8-fold), whereas the BIC for oritavancin was only halved. 

### 3.3. In Vivo G. mellonella Assays

Toxicity assays verified that neither fosfomycin or oritavancin alone, nor a combination of the two agents, was toxic to *G. mellonella* at the concentrations tested. The synergism observed between fosfomycin and oritavancin in time-kill assays was recapitulated in vivo using the *G. mellonella* infection model, with clear increases in the survival rates of larvae treated with a combination of fosfomycin and oritavancin compared to either agent alone.

As shown in Figure 3, in larvae infected with Ef-3, treatment with a combination of fosfomycin and oritavancin resulted in a higher survival rate (95%) relative to larvae treated with monotherapy (fosfomycin 70%, oritavancin 80%). Larvae treated with combination therapy exhibited significantly greater survival relative to larvae treated with fosfomycin alone (*p* = 0.037); however, this significance did not extend to larvae treated with oritavancin alone (*p* = 0.167). In the larvae infected with Ef-4, combination treatment of fosfomycin and oritavancin afforded significantly greater survival than either agent alone (*p* = 0.0038, *p* = 0.0005). Larvae infected with Ef-8 and treated with a combination of fosfomycin and oritavancin showed significantly greater survival relative to larvae treated with oritavancin (*p* = 0.0053) or fosfomycin (*p* = 0.0244) alone. Similarly, larvae infected with Ef-10 that were treated with fosfomycin and oritavancin in combination exhibited greater survival than those treated with fosfomycin only (*p* = 0.014) and oritavancin only (*p* = 0.013).

## 4. Discussion

Infections caused by VRE are an increasing problem worldwide. While good antibiotic options exist for infections other than BSI (e.g., liver abscesses, intra-abdominal infections), there are limited options for the treatment of BSI infections. The main drugs used against VRE are linezolid and daptomycin. Linezolid is a suboptimal drug for infections requiring a hydrophilic drug, and daptomycin can fail to reach adequate plasmatic concentrations, with consequent clinical failures. In theory, other antibiotics, such as chloramphenicol, eravacycline, tigecycline and tedizolid, would have anti-VRE activity, but clinical experience is limited and/or clinical breakpoints are lacking [18,19,20,21,22]. Oritavancin is a new antibiotic with excellent in vitro activity against VRE, but it is a highly lipophilic drug, therefore ill-suited for BSI. Practically, the combination of fosfomycin with oritavancin could bring several advantages, including the ability (1) to synergise with oritavancin, reducing MICs; (2) to provide a hydrophilic partner drug to treat intravascular infections with a better PK/PD profile; (3) to provide an advantage for biofilm-associated infections [23]. 

Other authors investigated the potential of oritavancin combinations against VRE. Smith et al. evaluated ceftaroline, ampicillin and ertapenem as partner drugs but they did not find a reliable synergism [24]. Wu et al. evaluated ceftriaxone, daptomycin, gentamicin and linezolid as partner drugs, with no synergy detected, apart from gentamicin, against strains not displaying high-level aminoglycoside resistance [25].

In the study presented here, the combination of oritavancin and fosfomycin increased drug susceptibility, demonstrating a synergistic effect in 80% of isolates and an additive effect in the remaining isolates. It is also noteworthy to highlight that the combination restored fosfomycin susceptibility in 85% of fosfomycin-resistant isolates. Our time-killing studies demonstrated that the combination of oritavancin and fosfomycin provided a CFU/mL reduction >2 log10 compared with the most effective drug alone; in addition, the combination prevented the bacterial regrowth seen after 8–24 h at sub-inhibitory drug concentrations. Preliminary investigations were also performed on the sessile cells embedded in the biofilm formed by two isolates, carrying a vanA and a vanB determinant, respectively. Both showed higher biofilm production when cultured in a blood-containing medium, which is of particular significance in infections such as endocarditis or cardiovascular device infection (pacemakers, aortic prosthesis, etc.). Finally, we demonstrated in vivo (*Galleria mellonella*) a higher survival rate of the combination therapy, compared to monotherapy (fosfomycin, oritavancin alone). 

## 5. Conclusions

We confirmed, in vitro and in vivo (*Galleria mellonella*), that oritavancin and fosfomycin display synergistic activity against VRE clinical strains, with preliminary data suggesting an increase in the antibacterial activity of these drugs when used in combination also against biofilm-producing isolates. Our study provides preclinical evidence to support trials combining oritavancin and fosfomycin for VRE BSI in humans, even when biofilms are involved.

## Figures and Tables

**Figure 1 antibiotics-11-01334-f001:**
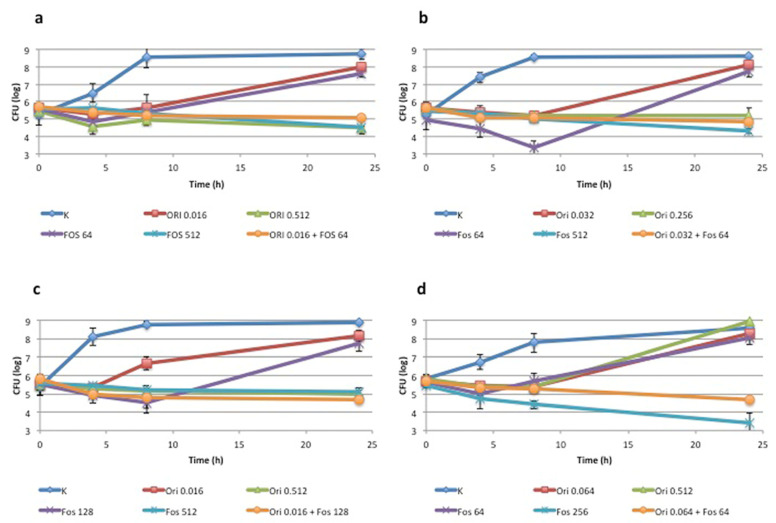
Time-kill curves showing log_10_ CFU/mL at different times (mean) for (**a**) Ef-3 (*vanB*), (**b**) Ef-4 (*vanB*), (**c**) Ef-8 (*vanA*) and (**d**) Ef-10 (*vanA*). Results are the means ± standard deviation of three independent experiments.

**Figure 2 antibiotics-11-01334-f002:**
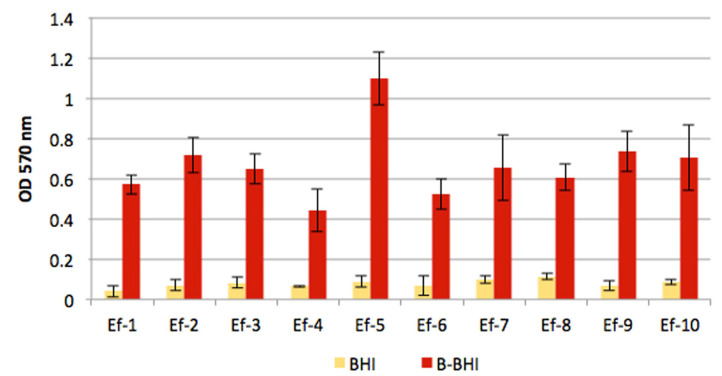
BF production by Ef isolates grown in BHI and in BHI supplemented with 5% defibrinated sheep blood (B-BHI). Each strain was always tested in triplicate and final results are the means of three independent experiments ± standard deviation.

**Figure 3 antibiotics-11-01334-f003:**
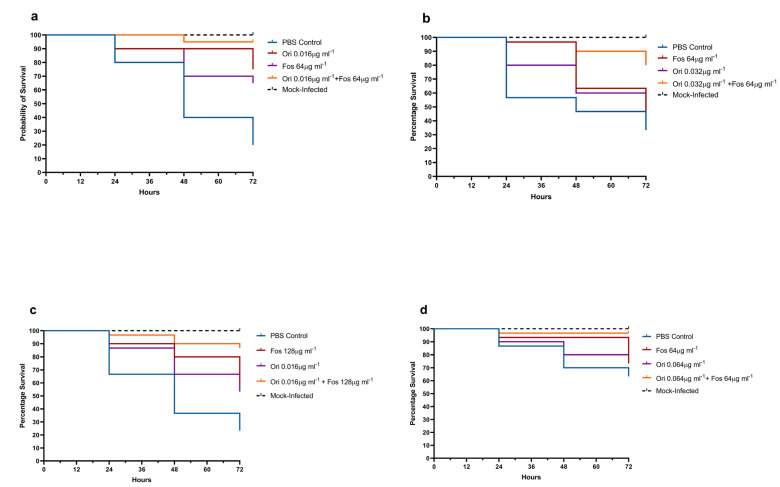
Survival rates for *Galleria mellonella* larvae treated with fosfomycin (Fos), oritavancin (Ori), or a combination of both antimicrobial agents. Larvae were infected with (**a**) Ef-3, (**b**) Ef-4, (**c**) EF-8, (**d**) Ef-10. Mock-infected controls refer to larvae that received a PBS injection in place of a bacterial strain. PBS controls refer to larvae that received bacterial inocula, but PBS in lieu of oritavancin, fosfomycin or combination treatment. Graphs show pooled data from three independent replicates consisting of 10 larvae.

**Table 1 antibiotics-11-01334-t001:** MIC values of fosfomycin (FOF) and oritavancin (ORI), alone and in combination, obtained by checkerboard assay, and the resulting FICI values (light green: synergistic effect; dark green: additive effect). Mode values of two independent experiments, carried out in triplicate, are reported.

	MIC µg/mL				
	Drugs Alone		Drugs in Combination		
	FOF	ORI	FOF	ORI	FICI
Ef-1 (*vanA*)	64	0.512	32	0.016	0.531
Ef-2 (*vanA*)	128	0.512	16	0.128	0.375
Ef-5 (*vanA*)	64	0.512	16	0.064	0.375
Ef-8 (*vanA*)	512	0.512	128	0.016	0.281
Ef-10 (*vanA*)	256	0.512	64	0.064	0.375
Ef-3 (*vanB*)	512	0.512	64	0.016	0.156
Ef-4 (*vanB*)	512	0.256	64	0.032	0.250
Ef-6 (*vanB*)	64	0.032	32	0.008	0.500
Ef-7 (*vanB*)	256	0.512	32	0.032	0.188
Ef-9 (*vanB*)	512	0.512	64	0.004	0.133

**Table 2 antibiotics-11-01334-t002:** BIC values of fosfomycin (FOF) and oritavancin (ORI), alone and in combination, on two representative isolates, and the resulting FICI values (light green: synergistic effect; dark green: additive effect)**.** Mode values of two independent experiments, carried out in triplicate, are reported.

	BIC µg/mL				
	Drugs Alone		Drugs in Combination		
	FOF	ORI	FOF	ORI	FICI
Ef-3 (*vanB*)	2048	8	64	0.5	0.094
Ef-10 (*vanA*)	>2048	2	256	1	0.563

## Data Availability

Not applicable.

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
