# Peer review of "In Vitro and In Vivo Studies of Oritavancin and Fosfomycin Synergism against Vancomycin-Resistant Enterococcus faecium"

_antibiotics, 2022, doi:10.3390/antibiotics11101334_

Round 1

Reviewer 1 Report

L44 needs references

L45 need to add some information on reason why biofilm is far more resistant than planktonic.

L64 Reference

L92 please explain in more detail how the combinational MIC was performed

L95 were experiment done twice in triplicate?

No statistical section in methods should be at the end, software used analysis etc. p significance  

 L192 Table 1 are the values mean? should add Standard Deviations, how many replicates

Table 1 legend what is shiny and shady green? What is meant be in combination – what drugs as they are listed as FOF and ORI. I thought if it is drugs in combination there would only be one MIC value for each strain.

L220 was there a statical reduction?

L229 should be under the figure not above.

L263 Was the experiment done in triplicate on three occasions? If so please state, move statement up to figure legend

L278 I am confused as to why there are two columns FOF and ORI under drugs in combination, should there be just one e.g., FOF + ORI, please state if done in triplicate and add mean and SD for each

L293 use same decimal points for all p values e.g. p = 0.037 and p = 0.167

Author Response

Reviewer 1

L44 needs references

Added

L45 need to add some information on reason why biofilm is far more resistant than planktonic.

Done

L64 Reference

Added

L92 please explain in more detail how the combinational MIC was performed

The checkerboard assay was explained in detail

L95 were experiment done twice in triplicate?

Yes it was. It has been added/specified.

No statistical section in methods should be at the end, software used analysis etc. p significance  

We prefer to maintain statistical methods separated, since every paragraph has its statistics.

L192 Table 1 are the values mean? should add Standard Deviations, how many replicates

A sentence was added to explain that MIC values are expressed as modal values (as currently used) and that only few results differed from the mode by no more than one doubling dilution (as currently accepted by both CLSI and EUCAST).

Table 1 legend what is shiny and shady green? What is meant be in combination – what drugs as they are listed as FOF and ORI. I thought if it is drugs in combination there would only be one MIC value for each strain.

We replaced shiny green with “light green” and shady green with “dark green”.

When you test two drugs “in combination”, you mix different concentrations of both of them and identify the well containing the lowest concentration of both drugs that inhibits bacterial growth (See figure attached)

So, MIC values “in combination” will be the concentration of each drug in that well. It cannot be a single value

- The meaning of FOF and ORI is specified in the legend

L220 was there a statical reduction?

For time-killing studies there are specific rules (Doern et al. 2014 – ref 15) that state to consider synergic an effect when the combination between 2 antibiotics determine a CFU reduction > 2 log10 compared to the most effective of them used alone. No other calculations/statistics are needed.

L229 should be under the figure not above.

Done

L263 Was the experiment done in triplicate on three occasions? If so please state, move statement up to figure legend

The experiment was actually performed in triplicate in three independent occasions, as we stated in the legend: “Each strain was always tested in triplicate and final results are the means of three independent experiments ± standard deviation.” However, there was a typo, because this sentence was repeated twice, so the second sentence was deleted.

L278 I am confused as to why there are two columns FOF and ORI under drugs in combination, should there be just one e.g., FOF + ORI, please state if done in triplicate and add mean and SD for each

Statement that experiments were repeated in triplicate was added. However, as explained above, results are expressed as modal values not as means ± standard deviation.

L293 use same decimal points for all p values e.g. p = 0.037 and p = 0.167

Done

Reviewer 2 Report

The article is interesting. It is well written and designed. The methods described are adequate for the tests performed in this study. However, there are few suggestions that I have, in order to improve the quality of this work:

- please, specify the identification method for the E.faecium strains used in the study (not only as a reference)

- please, specify the standard used for the initial interpretation of MIC values of the 10 VRE (E. faecium) strains

- was the quality control strain (E. faecalis) used only for the initial sensitivity testing or also in the sinergy testing?

- the Discussion section is slightly short, I would recommend to add more references regarding other therapeutic options for VRE.

Author Response

Reviewer 2

The article is interesting. It is well written and designed. The methods described are adequate for the tests performed in this study. However, there are few suggestions that I have, in order to improve the quality of this work:

- please, specify the identification method for the E. faecium strains used in the study (not only as a reference)

A brief description was added as required

- please, specify the standard used for the initial interpretation of MIC values of the 10 VRE (E. faecium) strains

It has been added: we used EUCAST breakpoints.

- was the quality control strain (E. faecalis) used only for the initial sensitivity testing or also in the sinergy testing?

In the synergy testing, it was used only with the two antibiotics alone to control their activity. It was not used with drugs in combination, because it was not considered useful in this case: nothing is known about its behaviour.   

- the Discussion section is slightly short, I would recommend to add more references regarding other therapeutic options for VRE.

Thank you. Done

Round 2

Reviewer 1 Report

I am happy with the revision